# Acute Kidney Injury in Patients Undergoing Total Hip Arthroplasty: A Systematic Review and Meta-Analysis

**DOI:** 10.3390/jcm8010066

**Published:** 2019-01-09

**Authors:** Charat Thongprayoon, Wisit Kaewput, Natanong Thamcharoen, Tarun Bathini, Kanramon Watthanasuntorn, Sohail Abdul Salim, Patompong Ungprasert, Ploypin Lertjitbanjong, Narothama Reddy Aeddula, Aldo Torres-Ortiz, Michael A. Mao, Wisit Cheungpasitporn

**Affiliations:** 1Division of Nephrology and Hypertension, Mayo Clinic, Rochester, MN 55905, USA; charat.thongprayoon@gmail.com (C.T.); mao.michael@mayo.edu (M.A.M.); 2Department of Military and Community Medicine, Phramongkutklao College of Medicine, Bangkok 10400, Thailand; wisitnephro@gmail.com; 3Division of Nephrology, Beth Israel Deaconess Medical Center, Harvard Medical School, Boston, MA 02215, USA; natthamcharoen@gmail.com; 4Department of Internal Medicine, University of Arizona, Tucson, AZ 85721, USA; tarunjacobb@gmail.com; 5Department of Internal Medicine, Bassett Medical Center, Cooperstown, NY 13326, USA; kanramon@gmail.com (K.W.); ploypinlert@gmail.com (P.L.); 6Division of Nephrology, Department of Medicine, University of Mississippi Medical Center, Jackson, MS 39216, USA; sohail3553@gmail.com (S.A.S.); Aldo_t86@hotmail.com (A.T.-O.); 7Clinical Epidemiology Unit, Department of Research and Development, Faculty of Medicine, Siriraj Hospital, Mahidol University, Bangkok 10700, Thailand; p.ungprasert@gmail.com; 8Division of Nephrology, Department of Medicine, Deaconess Health System, Evansville, IN 47747, USA; dr.anreddy@gmail.com

**Keywords:** acute kidney injury, acute renal failure, hip arthroplasty, hip Surgery, postoperative acute kidney injury, incidence, epidemiology, systematic reviews, meta-analysis

## Abstract

Background: The number of total hip arthroplasties (THA) performed across the world is growing rapidly. We performed this meta-analysis to evaluate the incidence of acute kidney injury (AKI) in patients undergoing THA. Methods: A literature search was performed using MEDLINE, EMBASE and Cochrane Database from inception until July 2018 to identify studies assessing the incidence of AKI (using standard AKI definitions of RIFLE, AKIN, and KDIGO classifications) in patients undergoing THA. We applied a random-effects model to estimate the incidence of AKI. The protocol for this meta-analysis is registered with PROSPERO (no. CRD42018101928). Results: Seventeen cohort studies with a total of 24,158 patients undergoing THA were enrolled. Overall, the pooled estimated incidence rates of AKI and severe AKI requiring dialysis following THA were 6.3% (95% CI: 3.8%–10.2%) and 0.5% (95% CI: 0.1%–2.3%). Subgroup analysis based on the countries by continent was performed and demonstrated the pooled estimated incidence of AKI following THA of 9.2% (95% CI: 5.6%–14.8%) in Asia, 8.1% (95% CI: 4.9%–13.2%) in Australia, 7.4% (95% CI: 3.2%–16.3%) in Europe, and 2.8% (95% CI: 1.2%–17.0%) in North America. Meta-regression of all included studies showed significant negative correlation between incidence of AKI following THA and study year (slope = −0.37, *p* <0.001). There was no publication bias as assessed by the funnel plot and Egger’s regression asymmetry test with *p* = 0.13 for the incidence of AKI in patients undergoing THA. Conclusion: The overall estimated incidence rates of AKI and severe AKI requiring dialysis in patients undergoing THA are 6.3% and 0.5%, respectively. There has been potential improvement in AKI incidence for patients undergoing THA over time.

## 1. Introduction

Acute kidney injury (AKI) is a complex clinical syndrome, characterized by an abrupt decrease in glomerular filtration, associated with various etiologies and pathophysiological pathways [1,2,3,4,5,6]. Globally, AKI is a common condition, affecting 13.3 million patients a year [3,4]. This complex syndrome is associated with significant morbidity, considerable mortality resulting in 1.7 million deaths a year, and increased hospital costs and subsequent burden on national health care budgets across the world [7,8].

Total hip arthroplasty (THA) is one of the most consistently successful and cost-effective orthopedic procedures performed today. It is indicated in patients with severe hip pain from a variety of conditions and ultimately provides significant improved pain relief, functionality and quality of life [9,10,11,12,13,14,15]. The total number of THA performed across the world is growing rapidly with close to 522,800 surgeries in the United States alone in 2014, making it the fourth most common operative procedure [16,17].

Previous studies have demonstrated different incidence of AKI following total joint arthroplasties using standard AKI criteria ranging from 0.5% to 22% based on types of joint arthroplasties [5,18,19,20,21,22,23,24,25,26,27,28,29,30,31,32,33,34,35,36,37,38,39,40]. The clinical and economic landscape of joint replacement surgeries including both general orthopedic and more specifically THA is rapidly changing, thus making it imperative for clinicians and administrators to understand the various risk factors and post-operative complications in order to better provide excellence in clinical and financial outcomes. However, despite progress in perioperative medicine, the incidence, incidence trend, and risk factors for AKI in patients following THA remain unclear [18,19,20,21,22,23,24,25,26,27,28,29,30,31,32,33,34,35,36,37,38,39,40]. In addition, complication rates such as reoperation and readmission among patients undergoing THA remain high [18,19,20,21,22,23,24,25,26,27,28,29,30,31,32,33]. Thus, we performed this systematic review to summarize the incidence and associated risk factors for AKI in patients undergoing THA.

## 2. Methods

### 2.1. Search Strategy and Literature Review

The protocol for this meta-analysis is registered with PROSPERO (International Prospective Register of Systematic Reviews; no. CRD42018101928). A systematic literature search of EMBASE (1988 to July 2018), MEDLINE (1946 to July 2018), and the Cochrane Database of Systematic Reviews (database inception to July 2018) was performed to evaluate the incidence of AKI in patients undergoing THA. The systematic literature review was undertaken independently by two investigators (C.T. and W.C) using the search strategy that combined the terms of “acute kidney injury” or “renal failure” and “hip arthroplasty” or “hip surgery” which is provided in Online Appendix A. No language limitation was applied. A manual search for conceivably relevant studies using references of the included articles was also performed. This study was conducted by the Preferred Reporting Items for Systematic Reviews and Meta-Analysis (PRISMA) statement [41] and the Strengthening the Reporting of Observational Studies in Epidemiology (STROBE) [42].

### 2.2. Selection Criteria

Eligible studies must be clinical trials or observational studies (cohort, case-control, or cross-sectional studies) that reported incidence of AKI in patients undergoing THA. Included studies must provide the data to estimate incidence of AKI with 95% confidence intervals (CI). Retrieved articles were individually reviewed for eligibility by the two investigators (C.T. and W.C.). Discrepancies were addressed and solved by mutual consensus. Inclusion was not limited by the size of the study.

### 2.3. Data Abstraction

A structured data collecting form was used to obtain the following information from each study: title, name of the first author, year of the study, publication year, country where the study was conducted, definition of THA, AKI definition, incidence of AKI, and risk factors for AKI.

### 2.4. Statistical Analysis

Analyses were performed utilizing the Comprehensive Meta-Analysis 3.3 software (Biostat Inc, Englewood, NJ, USA). Adjusted point estimates from each study were consolidated by the generic inverse variance approach of DerSimonian and Laird, which designated the weight of each study based on its variance [43]. Given the possibility of between-study variance, we used a random-effect model rather than a fixed-effect model. Cochran’s *Q* test and *I*^2^ statistic were applied to determine the between-study heterogeneity. A value of *I*^2^ of 0% to 25% represents insignificant heterogeneity, 26% to 50% low heterogeneity, 51% to 75% moderate heterogeneity and 76%–100% high heterogeneity [44]. The presence of publication bias was assessed by the Egger test [45].

## 3. Results

A total of 410 potentially eligible articles were identified using our search strategy. After the exclusion of 162 articles based on title and abstract for clearly not fulfilling inclusion criteria on the basis of type of article, study design, population or outcome of interest, and 209 due to being duplicates, 39 articles were left for full-length review. 11 of them were excluded from the full-length review as they did not report the outcome of interest while 5 articles were excluded because they were not observational studies. Six studies [9,10,11,12,13,14] were subsequently excluded because they did not use standard AKI definitions of Risk, Injury, Failure, Loss of kidney function, and End-stage kidney disease (RIFLE), Acute Kidney Injury Network (AKIN), and Kidney Disease: Improving Global Outcomes (KDIGO) classifications. Thus, 17 cohort studies [18,19,20,21,22,23,24,25,26,27,28,29,30,31,32,33,37] comprising 24,158 patients undergoing THA were included into the meta-analysis of associated AKI incidence. The literature retrieval, review, and selection process are demonstrated in Figure 1. The characteristics of the included studies are presented in Table 1.

### 3.1. Incidence of AKI in Patients Undergoing THA

Overall, the pooled estimated incidence rates of AKI and severe AKI requiring dialysis following THA were 6.3% (95% CI: 3.8%–10.2%, *I*^2^ = 97%, Figure 2A) and 0.5% (95% CI: 0.1%–2.3%, *I*^2^ = 42%, Figure 2B), respectively. Subgroup analysis based on AKI definitions was performed and showed the pooled estimated incidence rates of AKI of 5.5% (95% CI: 1.3%–21.0%, *I*^2^ = 98%) by RIFLE criteria, 8.4% (95% CI: 5.2%–13.3%, *I*^2^ = 95%) by AKIN criteria, and 4.2% (95% CI: 1.9%–9.0%, *I*^2^ = 96%) by KDIGO criteria, respectively. We conducted a sensitivity analysis by excluding with patients undergoing THA for hip fractures. The pooled estimated incidence rates of AKI following THA for non-fracture indications was 5.6% (95% CI: 3.3%–9.3%, *I*^2^ = 97%). The pooled estimated incidence rates of AKI following THA among patients with fractures was 14.2% (95% CI: 11.8%–16.9%, *I*^2^ = 0%).

Subgroup analysis based on the countries by continent was performed and demonstrated pooled estimated incidence of AKI following THA of 9.2% (95% CI: 5.6%–14.8%) in Asia, 8.1% (95% CI: 4.9%–13.2%) in Australia, 7.4% (95% CI: 3.2%–16.3%) in Europe, and 2.8% (95% CI: 1.2%–17.0%) in North America.

Meta-regression of all included studies showed significant negative correlation between incidence of AKI following THA and the study year (slope = −0.37, *p* <0.001), as shown in Figure 3.

### 3.2. Risk Factors for AKI in Patients Undergoing THA

Reported risk factors for AKI in patients undergoing THA are demonstrated in Table 2 [18,19,20,22,23,24,26,27,29,30,31,32,33,46,47,48,49,50,51,52,53,54,55,56,57,58,59,60,61,62]. Older age [23,24,26,29,31,55,60], higher body mass index (BMI) [19,22,23,24,26,30,31,46,47,48,49,56,57,59], reduced baseline estimated glomerular filtration rate (eGFR)/chronic kidney disease (CKD) [31,54,55,57,60], diabetes mellitus (DM) [24,26,31,34,47,55,57], nonsteroidal anti-inflammatory drug (NSAID) [23,32,53] use, and perioperative blood transfusion [23,47,53,55] were consistently identified as important risk factors for AKI in patients undergoing THA.

### 3.3. Evaluation for Publication Bias

Funnel plot (Appendix A) and Egger’s regression asymmetry test were performed to evaluate for publication bias in analysis evaluating incidence of AKI in patients undergoing THA. There was no significant publication bias in meta-analysis assessing incidence of AKI in patients undergoing THA, *p* = 0.20.

## 4. Discussion

In this systematic review and meta-analysis, we found that patients who underwent THA had incidences of AKI (using a standard AKI definition) and AKI requiring dialysis of 6.3% and 0.5%, respectively. In addition, our findings showed a statistically significant negative correlation between incidence of AKI following THA and the study year, representing potential improvement in the AKI incidence for patients undergoing THA over time.

Similar to other types of perioperative AKI, the pathogenesis of THA-related AKI is multifactorial including intraoperative hypotension, perioperative anemia and blood transfusion, antibiotic-related AKI, and nephrotoxic agents such as NSAID use [18,19,20,22,23,24,26,27,29,30,31,32,33,46,47,48,49,50,51,52,53,54,55,56,57,58,59,60,61,62]. Our study demonstrated an overall low incidence of AKI following THA of 6.3%, and comparatively much lower than AKI incidence following other types of major surgeries such as coronary artery bypass grafting (AKI incidence approximately 20%) [63,64], vascular surgery (AKI incidence approximately 25%) [35,65], or heart valve replacement surgery (AKI incidence approximately 35% to 47%) [66]. In addition, the majority of AKI following THA is mild in severity [18,19,20,21,22,23,24,25,26,27,28,29,30,31,32,33] and only 0.5% of patients undergoing THA developed severe AKI requiring dialysis, as demonstrated by our meta-analysis. Despite the low incidence of AKI among patients undergoing THA, those who develop AKI following THA still carry significant increased mortality and worse outcomes including prolonged hospital stay [39,46,67,68] and hospital readmissions after THA [69,70]. In addition, it is now evident that even mild AKI is still associated with poor long-term outcomes including development of cardiovascular diseases and CKD [71,72]. Thus, perioperative medicine remains critically important in patients undergoing THA in order to prevent perioperative AKI [73].

Our study also showed a potential improvement in the AKI incidence following THA over time. This is likely explained by advances in perioperative medicine care [73]. As described earlier in this systematic review, the pathogenesis of AKI in patients undergoing THA is multifactorial. Risk factors for AKI following THA are summarized in Table 2. Since older age, higher BMI, CKD, DM were consistently identified as important risk factors for AKI in patients undergoing THA [18,19,20,22,23,24,26,27,29,30,31,32,33,46,47,48,49,50,51,52,53,54,55,56,57,58,59,60,61,62], clinicians and nephrologists should pay close attention to particular populations of patients. Limitation of nephrotoxic agents such as NSAIDs and a judicious use of perioperative blood transfusion should be considered [23,47,53,55] for preventive measures to reduce AKI following THA. In patients undergoing THA, recent studies have demonstrated significant avoidance of NSAID use in those possessing higher AKI risk, such as patients with CKD and congestive heart failure (CHF) [29,31]. In addition, in recent years, the use of hydroxyethyl starch (HES) solutions, unnecessary blood transfusions, chloride-rich intravenous fluids, and aminoglycosides have been discouraged [29,52,73]. Furthermore, although the use of angiotensin-converting enzyme inhibitors/angiotensin receptor blockers (ACEI/ARB) and perioperative AKI risk remains controversial [2,74,75], ACEIs/ARBs have been commonly discontinued before THA to prevent intraoperative hypotension [29]. Future studies are required to assess if discontinuation of ACEIs/ARBs before THA may affect the incidence of THA-associated AKI, and whether it should be included in AKI preventative strategies and care optimization among patients undergoing THA [37,40].

Several limitations in our meta-analysis are worth mentioning. First, there are statistical heterogeneities in our study. Possible sources for heterogeneities were the differences in patient characteristics between the individual studies. Studies involving large sample size observed lower AKI incidence compared to those with small sample size. Studies outside USA observed higher AKI incidence following THA, suggesting potential differences in operative surgery, antibiotic and blood transfusion, and perioperative care. Subgroup analysis based on the countries by continent demonstrated significantly different AKI incidences in each continent. Furthermore, the meta-regression analysis that demonstrated a significant negative correlation of post-THA AKI incidence with time adds the contribution of study year as a source of heterogeneity in our study. Second, there was limited data on the AKI incidence following THA from countries in South America and Africa. Third, there is a lack of data from the included studies on novel AKI biomarkers and AKI diagnosis based on urine output criteria. Lastly, this is a systematic review and meta-analysis of cohort studies and the data from population based studies were limited. Thus, future population based studies evaluating the incidence of AKI following THA are required.

In summary, there is an overall low incidence of AKI of 6.3% among patients undergoing THA. There has also been potential improvement in AKI incidence for patients undergoing THA over time.

## Figures and Tables

**Figure 1 jcm-08-00066-f001:**
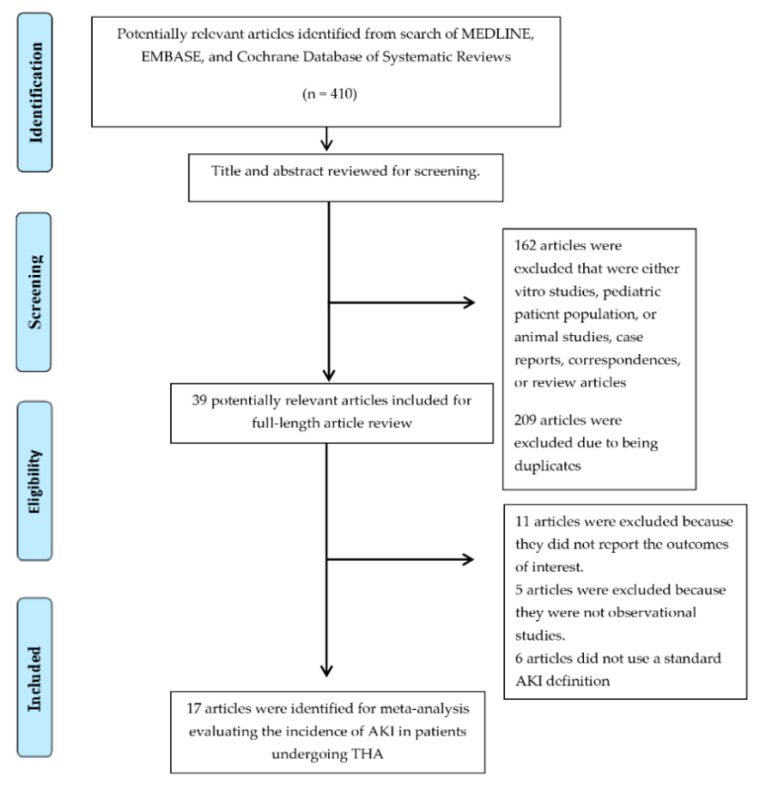
Outline of our search methodology.

**Figure 2 jcm-08-00066-f002:**
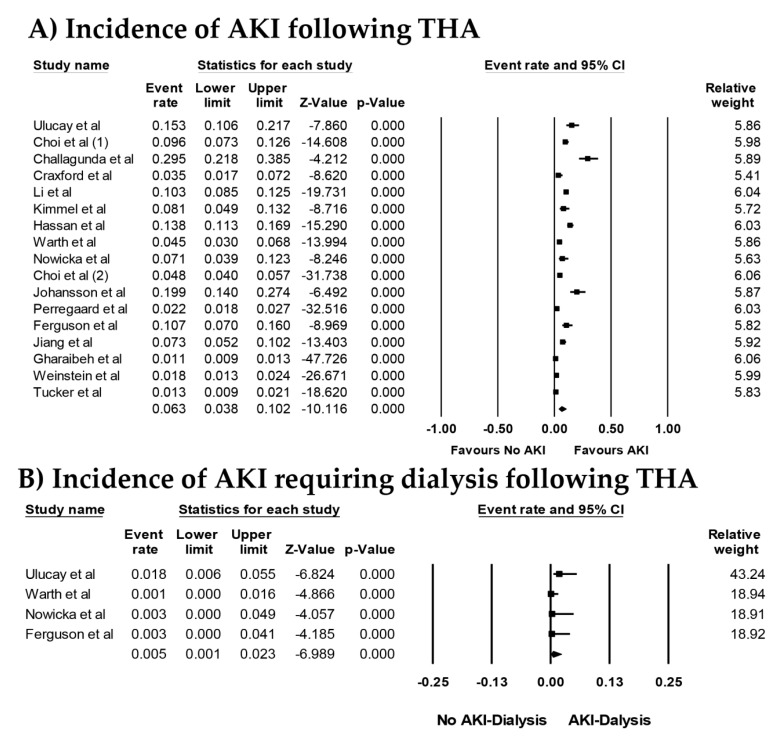
Forest plots of the included studies [18,19,20,21,22,23,24,25,26,27,28,29,30,31,32,33] assessing (**A**) incidence rates of AKI and (**B**) incidence rates of AKI requiring dialysis following THA. A diamond data marker represents the overall rate from each included study (square data marker) and 95% confidence interval (CI).

**Figure 3 jcm-08-00066-f003:**
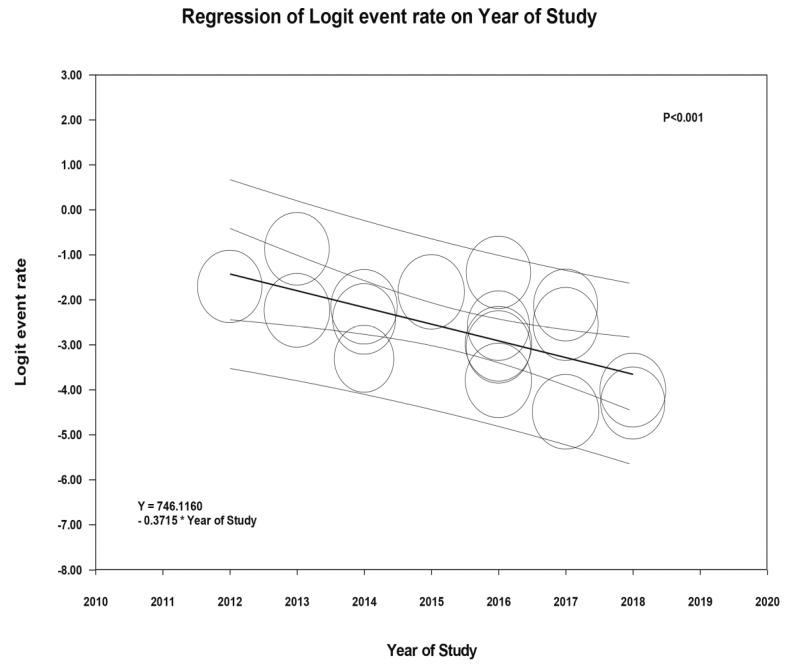
Meta-regression analyses showed significant negative correlation between incidence of AKI following THA and the study year (slope = −0.37, *p* <0.001). The solid black line represents the weighted regression line based on variance-weighted least squares. The inner and outer lines show the 95% confidence interval and prediction interval around the regression line. The circles indicate log event rates in each study.

**Table 1 jcm-08-00066-t001:** Main characteristic of studies included in meta-analysis of AKI incidence in patients undergoing THA [18,19,20,21,22,23,24,25,26,27,28,29,30,31,32,33].

Study	Year	Country	Study Design	Procedures/Patients	Number of Patients	AKI Definition	Incidence
Ulucay et al. [18]	2012	Turkey	Cohort	Cemented bipolar hip arthroplasty for femur neck fracture; aged ≥65 years	163	Acute kidney injury; AKIN criteria	25/163 (15.3%) Dialysis 3/163 (1.8%)
Choi et al. [1,19]	2013	South Korea	Cohort	Total hip replacement for avascular necrosis	489	Acute kidney injury; AKIN criteria	47/489 (9.6%)
Challagunda et al. [20]	2013	UK	Cohort	Elective hip surgery	112	Acute kidney injury; RIFLE criteria	33/112 (29.5%)
Craxford et al. [21]	2014	UK	Cohort	Total hip replacement	200	Acute kidney injury; RIFLE criteria	7/200 (3.5%)
Li et al. [22]	2014	China	Cohort	Total hip replacement	900	Acute kidney injury; AKIN criteria	93/900 (10.3%)
Kimmel et al. [23]	2014	Australia	Cohort	Primary elective total hip arthroplasty	173	Acute kidney injury; RIFLE criteria	14/173 (8.1%)
Hassan et al. [37]	2015	Denmark	Cohort	Primary hip replacement	586	Acute kidney injury; RIFLE criteria	81/586 (13.8%)
Warth et al. [24]	2016	USA	Cohort	Total hip arthroplasty	488	Acute kidney injury; AKIN criteria	22/488 (4.5%) Dialysis 0/488 (0%)
Nowicka et al. [25]	2016	UK	Cohort	Elective hip arthroplasty	156	Acute kidney injury; AKIN criteria	11/156 (7.1%) Dialysis 0/156 (0%)
Choi et al. [2,26]	2016	South Korea	Cohort	Total hip replacement	2467	Acute kidney injury; AKIN criteria	119/2467 (4.8%)
Johansson et al. [27]	2016	Denmark	Cohort	Elective total hip replacement	136	Acute kidney injury; KDIGO criteria	27/136 (19.9%)
Perregaard et al. [28]	2016	Denmark	Cohort	Primary elective total hip replacement	3416	Acute kidney injury; KDIGO criteria	75/3416 (2.2%)
Ferguson et al. [29]	2017	UK	Cohort	Primary hip arthroplasty	187	Acute kidney injury; KDIGO criteria	20/187 (10.7%) Dialysis 0/187 (0%)
Jiang et al. [30]	2017	USA	Cohort	Total hip arthroplasty	411	Acute kidney injury; RIFLE criteria	30/411 (7.3%)
Gharaibeh et al. [31]	2017	USA	Cohort	Total hip arthroplasty	10323	Acute kidney injury; KDIGO criteria	114/10323 (1.1%)
Weinstein et al. [32]	2018	USA	Cohort	Unilateral total hip arthroplasty with intraoperative controlled hypotension under neuraxial anesthesia	2431	Acute kidney injury; RIFLE or AKIN criteria	45/2431 (1.9%)
Tucker et al. [33]	2018	UK	Cohort	Primary hip arthroplasty	1420	Acute kidney injury; AKIN criteria	19/1420 (1.3%)

Abbreviations: AKI, acute kidney injury; AKIN, Acute Kidney Injury Network;KDIGO, Kidney Disease Improving Global Outcomes; RIFLE, Risk, Injury, Failure, Loss of kidney function, and End-stage kidney disease; THA, total hip arthroplasties; UK, United Kingdom; USA, United States of America.

**Table 2 jcm-08-00066-t002:** Reported Risk Factors for AKI in patients undergoing THA [18,19,20,22,23,24,26,27,29,30,31,32,33,46,47,48,49,50,51,52,53,54,55,56,57,58,59,60,61,62].

Study	Risk Factors for AKI
Jafari et al. [46]	Elevated BMI, elevated baseline creatinine, history of COPD, liver disease, CHF, hypertension, underlying heart disease
Weingarten et al. [47]	Elevated BMI, DM, the number of baseline antihypertensive medications, cerebral or peripheral vascular disease, the use of general anesthesia, perioperative blood transfusion
Ulucay et al. [18]	Lower baseline eGFR
Choi et al. [1,19]	Transplantation, increased weight
Li et al. [22]	Transplantation, increased weight
Challagundla et al. [20]	Male, ACEI/ARB
Nielson et al. [48]	Preoperative ACEI/ARB, BMI, CAD, intra-operative hypotension
Kimmel et al. [23]	Older age, increased BMI, lower baseline eGFR, NSAID use, ACEI/ARB use, blood transfusion
Ward et al. [49]	Increased BMI
Marty et al. [50]	Postoperative resistive index
Courtney et al. [51]	Dual antibiotics prophylaxis (cefazolin + vancomycin vs. cefazolin), ASA classification, preoperative kidney disease
Opperer et al. [52]	Perioperative fluid resuscitation with 6% HES or 5% albumin
Aeng et al. [53]	Gentamicin in premanufactured bone cement, intraoperative blood transfusion, postoperative NSAID use
Warth et al. [24]	Older age, elevated BMI, DM, smoking
Tan et al. [54]	CKD
Nadkarni et al. [55]	Older age, male sex, black race, CKD, CHF, chronic liver disease, hypertension, DM, atrial fibrillation, HCV infection, postoperative sepsis, acute MI, blood transfusion, urban hospital, small hospital size
Meller et al. [56]	Morbid obesity
Choi et al. [2,26]	Postoperative anemia, older age, male sex, BMI <22 or ≥25, DM, beta-blocker, ARB use
Ferguson et al. [29]	Older age, the use of ≥1 L of postoperative fluid
Johansson et al. [27]	Gentamicin, female sex
Geller et al. [57]	Higher BMI, lower baseline hemoglobin, history of comorbid condition (DM, CKD, CVD, hypertension)
Jiang et al. [30]	Perioperative ACEI/ARB use, vancomycin use, increased BMI
Zainudheen et al. [58]	Use of renin-angiotensin antagonists
Jamsa et al. [59]	Lower preoperative eGFR, ASA classification, BMI, duration of operation
Ghareibeh et al. [31]	Entire cohort: older age, male, CKD, heart failure, diabetes, hypertension Nested case control: elevated BMI, heart failure, DM, hypertension, lower GFR, transfusion
Yadav et al. [60]	Older age, surgery for periprosthetic joint infection, CKD, total number of surgeries
Weinstein et al. [32]	Lower baseline GFR, lower baseline hemoglobin, previous NSAID use
Tucker et al. [33]	Gentamicin use
Klement et al. [61]	Co-occurrence of a mental illness and a substance abuse disorder
Abar et al. [62]	Elevated preoperative creatinine, larger postoperative drop in hemoglobin, and higher ASA classification
Hassan et al. [37]	Older age, hypertension, general anesthesia, high ASA score, low baseline systolic and diastolic blood pressure, hip fracture
Dubrovskaya et al. [34]	Hospital stay >1 day prior to surgery, knee or hip surgery, DM
Bailey et al. [38]	Prophylactic use of flucloxacillin and gentamicin versus cefuroxime alone

Abbreviations: ACEI/ARB, angiotensin-converting enzyme inhibitor/angiotensin II receptor blockers; AKI, acute kidney injury; ASA, American Society of Anesthesiologists; BMI, body mass index; CAD, coronary artery disease; CHF, congestive heart failure; CKD, chronic kidney disease; COPD, chronic obstructive pulmonary disease; CVD, cardiovascular disease; DM, diabetes mellitus; eGFR, estimated glomerular filtration rate; HCV, hepatitis C virus; NSAID, nonsteroidal anti-inflammatory drug; THA, total hip arthroplasties.

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
