# Peer review of "Acute Kidney Injury in Patients Undergoing Total Hip Arthroplasty: A Systematic Review and Meta-Analysis"

_jcm, 2019, doi:10.3390/jcm8010066_

Reviewer 1 Report

Review comment sheet
1 A brief summary
   The paper describes that there is an overall low incidence of AKI of 6.3% among patients undergoing THA. There has also been potential improvement in AKI incidence for patients undergoing THA over time.

   This is a nice paper. However, I have some comments.
2 Overall evaluation
   The findings from this paper are excellent and novel regardless of a review.
3 Main problem
   This manuscript contained some questions described below.
   I think this paper has very interesting, this study contributes to future's clinical medicine largely. I have some questions from a point of view of clinical medicine.

   In this paper the authors focused the AKI after THA, I would like to know the other orthopedics operation, for example arm, shoulder, knee, backbone. Why do the authors focused the THA ?

   I want to know that what is the reason of indication of THA. I think the reason is fracture of hip joint, but the importance is that why the patients break the bones ? I think the background are high age, low bone mineral density, CKD and so on. And concerning to the relation between patient’s background and incidence of AKI, I want to know about that.

   I want to know the treatment of AKI after THA. Are there any distinctive therapy ? And when and how the nephrologists involved the treatment of AKI ?

Author Response

Point-by-point response to reviewers’ comments:
Reviewer #1
General comment

The paper describes that there is an overall low incidence of AKI of 6.3% among patients undergoing THA. There has also been potential improvement in AKI incidence for patients undergoing THA over time.

This is a nice paper. However, I have some comments.

The findings from this paper are excellent and novel regardless of a review.

       Response: We thank you for reviewing our manuscript and for your critical evaluation. We really appreciated your input and found your suggestions very helpful.
Comment#1 This manuscript contained some questions described below.

   I think this paper has very interesting, this study contributes to future's clinical medicine largely. I have some questions from a point of view of clinical medicine.

  In this paper the authors focused the AKI after THA, I would like to know the other orthopedics operation, for example arm, shoulder, knee, backbone. Why do the authors focused the THA ?

Response: We agree and appreciated the reviewer’s important input. We agree that the other orthopedics operations are also important. Since acute kidney injury is a heterogeneous condition consisting of distinct phenotypes based on its etiology and prognosis, and type of operations. Previous literature has shown incidence of AKI following total joint arthroplasties using standard acute kidney injury criteria ranging from 0.5% to 22% based on types of joint arthroplasties. Since studies have shown high complication risks such as reoperation and readmission among patients undergoing THA, we conducted this study to summarize the incidence and associated risk factors for AKI in patients undergoing THA. We appreciated the reviewer’s important input and we added the important point as the reviewer’s suggestion. The following text has been changed in bold in the introduction section:

Previous studies have demonstrated different incidence of AKI following total joint arthroplasties using standard AKI criteria ranging from 0.5% to 22% based on types of joint arthroplasties (5, 18-40). The clinical and economic landscape of joint replacement surgeries including both general orthopedic and more specifically THA is rapidly changing, thus making it imperative for clinicians and administrators to understand the various risk factors and post-operative complications in order to better provide excellence in clinical and financial outcomes. However, despite progress in perioperative medicine, the incidence, incidence trend, and risk factors for AKI in patients following THA remain unclear (18-40). In addition, complication rates such as reoperation and readmission among patients undergoing THA remain high (18-33). Thus, we performed this systematic review to summarize the incidence and associated risk factors for AKI in patients undergoing THA.” 

Comment#2 I want to know that what is the reason of indication of THA. I think the reason is fracture of hip joint, but the importance is that why the patients break the bones ? I think the background are high age, low bone mineral density, CKD and so on. And concerning to the relation between patient’s background and incidence of AKI, I want to know about that.

Response: Thank you for your suggestion. The reviewer raised very important point regarding the indication for THA may affect the incidence of AKI following THA. We carefully reviewed all included studies again and performed additional analysis. We found that those who underwent THA for fracture hip had higher incidence of AKI than those who had THA for other indications such as AVN or osteoarthritis. We appreciated the reviewer’s important input and we added the important point as the reviewer’s suggestion. The following text has been added in bold in the result section:

“Overall, the pooled estimated incidence rates of AKI and severe AKI requiring dialysis following THA were 6.3% (95%CI: 3.8%-10.2%, I2 = 97%, Figure 2A) and 0.5% (95%CI: 0.1%-2.3%, I2 = 42%, Figure 2B), respectively. We conducted a sensitivity analysis by excluding with patients undergoing THA for hip fractures. The pooled estimated incidence rates of AKI following THA for non-fracture indications was 5.6% (95%CI: 3.3%-9.3%, I2 = 97%). The pooled estimated incidence rates of AKI following THA among patients with fractures was 14.2% (95%CI: 11.8%-16.9%, I2 = 0%).”

Comment#3 I want to know the treatment of AKI after THA. Are there any distinctive therapy ? And when and how the nephrologists involved the treatment of AKI ?

Response: The reviewer’s point is well taken. While AKI is a significant cause of morbidity and mortality among hospitalization, there are currently no targeted therapies available. While waiting for future studies to identify targeted therapies or to promote renal recovery following AKI, perioperative medicine remains critically important in patients undergoing THA in order to prevent perioperative AKI. While there is no consensus on when the nephrologists should involve in perioperative management for patients undergoing THA, we agree with the reviewer on this important point and thus added more discussion on Nephrologist's perspective of perioperative management for patients undergoing THA. The following text has been changed in bold in the discussion section as the reviewer’s suggestion:

As described earlier in this systematic review, the pathogenesis of AKI in patients undergoing THA is multifactorial. Risk factors for AKI following THA are summarized in Table 2. Since older age, higher BMI, CKD, DM were consistently identified as important risk factors for AKI in patients undergoing THA (18-20, 22-24, 26, 27, 29-33, 46-62), clinicians and nephrologists should pay close attention to particular populations of patients. Limitation of nephrotoxic agents such as NSAIDs and a judicious use of perioperative blood transfusion should be considered (23, 47, 53, 55) for preventive measures to reduce AKI following THA. In patients undergoing THA, recent studies have demonstrated significant avoidance of NSAID use in those possessing higher AKI risk, such as patients with CKD and congestive heart failure (CHF) (29, 31). In addition, in recent years, the use of hydroxyethyl starch (HES) solutions, unnecessary blood transfusions, chloride-rich intravenous fluids, and aminoglycosides have been discouraged (29, 52, 73). Furthermore, although the use of angiotensin-converting enzyme inhibitors/angiotensin receptor blockers (ACEI/ARB) and perioperative AKI risk remains controversial (2, 74, 75), ACEIs/ARBs have been commonly discontinued before THA to prevent intraoperative hypotension (29). Future studies are required to assess if discontinuation of ACEIs/ARBs before THA may affect the incidence of THA-associated AKI, and whether it should be included in AKI preventative strategies and care optimization among patients undergoing THA (37, 40).”

All authors thank the Editors and reviewers for their valuable suggestions. The manuscript has been improved considerably by the suggested revisions! 

Reviewer 2 Report

In this study, authors presented a systematic review and metanalysis of incidence of kidney injury following hip replacement. Authors found low incidence of AKI (6%) and AKI requiring dialysis (0.5%) and AKI incidence decreased over years following THA. Overall, data and method are well presented, however as pointed out by authors in limitation section about sample heterogeneity which could bring variation.

Studies involving large sample size observed lower AKI incidence compared to small sample size.

Studies outside US observed higher AKI incidence following THA suggesting differences in operative surgery, antibiotic and blood transfusion quality and procedures.

Did authors adjust data with confounding variables such as age, sex, chronic illness etc?

Author Response

Reviewer #2
In this study, authors presented a systematic review and meta-analysis of incidence of kidney injury following hip replacement. Authors found low incidence of AKI (6%) and AKI requiring dialysis (0.5%) and AKI incidence decreased over years following THA. Overall, data and method are well presented, however as pointed out by authors in limitation section about sample heterogeneity which could bring variation.

Response: We thank you for reviewing our manuscript and for your critical evaluation. We really appreciated your input and found your suggestions very helpful.

Comment#1 Studies involving large sample size observed lower AKI incidence compared to small sample size. Studies outside US observed higher AKI incidence following THA suggesting differences in operative surgery, antibiotic and blood transfusion quality and procedures.

       Response: We appreciated the reviewer’s important input and observation. The reviewer raised very important point that differences in operative surgery, antibiotic and blood transfusion quality and procedures may have played important impacts on heterogeneity of our meta-analysis. We appreciated the reviewer’s important input and we added the important point as the reviewer’s suggestion. The following text has been added in bold in the discussion/limitation section:

“Several limitations in our meta-analysis are worth mentioning. First, there are statistical heterogeneities in our study. Possible sources for heterogeneities were the differences in patient characteristics between the individual studies. Studies involving large sample size observed lower AKI incidence compared to those with small sample size. Studies outside USA observed higher AKI incidence following THA, suggesting potential differences in operative surgery, antibiotic and blood transfusion, and perioperative care.”
Comment#2 Did authors adjust data with confounding variables such as age, sex, chronic illness etc?

Response: We agree with the reviewer that there are many factors that affect the risk of AKI following THA. We agree with these important factors and we have summarized the reported predictors/associated factors for AKI following THA in Table 2. In our meta-analysis, we assessed the incidence of AKI in patients undergoing THA and thus summarized the incidence of AKI following THA, which could not adjust these variables. We respected the reviewer’s important comment. Based on available data from included studies, we additionally performed additional analysis based on the indications for THA which may have affected the AKI risk. We found that those who underwent THA for fracture hip had higher incidence of AKI than those who had THA for other indications such as AVN or osteoarthritis. We appreciated the reviewer’s important input and we added the important point as the reviewer’s suggestion. The following text has been added in bold in the result section:

“Overall, the pooled estimated incidence rates of AKI and severe AKI requiring dialysis following THA were 6.3% (95%CI: 3.8%-10.2%, I2 = 97%, Figure 2A) and 0.5% (95%CI: 0.1%-2.3%, I2 = 42%, Figure 2B), respectively. We conducted a sensitivity analysis by excluding with patients undergoing THA for hip fractures. The pooled estimated incidence rates of AKI following THA for non-fracture indications was 5.6% (95%CI: 3.3%-9.3%, I2 = 97%). The pooled estimated incidence rates of AKI following THA among patients with fractures was 14.2% (95%CI: 11.8%-16.9%, I2 = 0%).”

All authors thank the Editors and reviewers for their valuable suggestions. The manuscript has been improved considerably by the suggested revisions! 

Round  2

Reviewer 1 Report

In this paper, answers are properly and carefully described in my question, and it has been corrected well.

Author Response

Thank you.

Reviewer 2 Report

Authors have satisfactorily answered my comments, I have no further questions.

Author Response

Thank you.